# A novel multi-model feature generation technique for suicide detection

Ting Ding[1,2,*], Tonghui Qu[3,*], Zongliang Zou[1] and Cheng Ding[4]

[1] School of Earth Science, East China University of Technology, Nanchang, Jiangxi, China
[2] Urumqi Comprehensive Survey Center on Natural Resources, China Geological Survey, Urumqi, Xinjiang, China
[3] Hangzhou Hikvision Digital Technology, Hangzhou, China
[4] Department of Biomedical Engineering, Emory University, Atlanta, GA, United States of America
[*] These authors contributed equally to this work.

Corresponding author
Cheng Ding, cheng.ding2@emory.edu

## ABSTRACT

Automated expert systems (AES) analyzing depression-related content on social media have piqued the interest of researchers. Depression, often linked to suicide, requires early prediction for potential life-saving interventions. In the conventional approach, psychologists conduct patient interviews or administer questionnaires to assess depression levels. However, this traditional method is plagued by limitations. Patients might not feel comfortable disclosing their true feelings to psychologists, and counselors may struggle to accurately predict situations due to limited data. In this context, social media emerges as a potentially valuable resource. Given the widespread use of social media in daily life, individuals often express their nature and mental state through their online posts. AES can efficiently analyze vast amounts of social media content to predict depression levels in individuals at an early stage. This study contributes to this endeavor by proposing an innovative approach for predicting suicide risks using social media content and machine learning techniques. A novel multi-model feature generation technique is employed to enhance the performance of machine learning models. This technique involves the use of a feature extraction method known as term frequency-inverse document frequency (TF-IDF), combined with two machine learning models: logistic regression (LR) and support vector machine (SVM). The proposed technique calculates probabilities for each sample in the dataset, resulting in a new feature set referred to as the probability-based feature set (ProBFS). This ProBFS is compact yet highly correlated with the target classes in the dataset. The utilization of concise and correlated features yields significant outcomes. The SVM model achieves an impressive accuracy score of 0.96 using ProBFS while maintaining a low computational time of 5.63 seconds even when dealing with extensive datasets. Furthermore, a comparison with state-of-the-art approaches is conducted to demonstrate the significance of the proposed method.

# INTRODUCTION

Depression can cause suicidal thoughts in some sufferers (*Sharma & Morishetty, 2023*; *Dong & Yang, 2021*). More than 300 million people are affected by depression, as reported

by the World Health Organization (WHO) (*Uddin et al., 2022*). Work overload, family issues, and income problems can be the causes of depression, and it can severely affect workplaces, family affairs, and educational institutes. Patients with depression can harm themselves and sometimes others, leading to personal or public injuries (*Calagua-Bedoya et al., 2023*). Depression is also a significant leading cause of non-fatal health issues, and it is more common in low-income or developing countries. Now, it is also affecting people from all around the world, and its prevalence is increasing day by day. According to a WHO report, the percentage of depression cases increased by 18.4% between 2005 to 2015 (*World Health Organization, 2017*). Depression can lead to suicide, and according to a report by WHO, over 800,000 suicide deaths occur every year. Every 40 s, a person commits suicide, and the leading cause is depression. This suicide rate has increased by 3.7% in recent years (*Burdisso, Errecalde & Montes-y Gómez, 2019*).

Early prediction of depression levels can be helpful in saving lives and can also reduce its rapidly growing prevalence. Early diagnosis of depression can also help minimize the negative impact on an individual's social, economic, and personal life, as well as on the community (*Burdisso, Errecalde & Montes-y Gómez, 2019*). Many expert systems are available in the healthcare domain for early disease diagnosis, but there is a lack of expert systems for suicide detection. Most researchers have used personal descriptive methods or textual data for identifying depression. Textual data, such as social media posts, can be utilized to detect a patient's depression state, while descriptive methods involve the use of personal information such as gender, income, drug consumption, age, and other related factors (*Fujita et al., 2021*).

Traditional approaches to depression prediction are very simple. Most psychologists prefer to have conversations with the patient and based on conversation points they predict the depression level. The psychologist also used surveys or questionnaires methods to identify the stress. They give questionnaires paper to the patient to fill with answers. They used this text data to predict the depression status of a patient (*Goyal et al., 2016*). Previous psychological researchers revealed that textual data can be good for predicting depression and they find links between a user's social media posts and his depression situation (*Ortega-Mendoza et al., 2022*). Psychologists examine the person's social media activity which is available online and easily available to the psychologist to predict the patient's situation. With this trend, the researcher concluded that online data can be significant in developing an automated expert system (AES) for decision-making on depression identification.

The AES can identify depression early which ensures the proper treatment of the patient. Since all information is publically available on a person's social media platform such as Twitter, Facebook, and Instagram AES can easily analyze a person's situation using that data. Some previous studies also highlight how text reveal someone's feeling, thoughts, emotion, *etc* (*Schwartz & Ungar, 2015*). For example, *Kmetty & Bozsonyi (2022)* used Facebook activities to identify the depression-related behavior of a person. In this way, there is no need for surveys or questionnaires which can collect only limited data. To construct the AES for depression prediction NLP and machine learning techniques can be adopted to achieve efficient results using text data.

Some researchers have proposed using AES for depression identification. For instance, in *Naseem et al. (2022a)*, NLP techniques were employed to propose an early depression detection system. They utilized support vector machine (SVM) and neural networks (NN) models to train on text data and make predictions. Similarly, in *Sharma & Neema, (2023)*, social media text posts were used to detect depression, and long short-term memory (LSTM) and recurrent neural networks (RNN) models were deployed. However, previous studies often grappled with challenges related to small or imbalanced datasets. It is also important to note that these studies primarily focused on contributions at the classification level, but it is equally valuable to make contributions at the feature engineering level. Such contributions can significantly improve classification results. In response to these limitations, our study harnesses a considerably larger dataset. Furthermore, while there is room for improvement in both accuracy and efficiency in existing methodologies, our study introduces a pioneering feature generation approach. Our feature engineering approach, which combines LR and SVM probabilities, can lead to more informative features. This enhanced discrimination can be particularly useful in situations where the distinction between classes is subtle. This approach not only promises remarkable accuracy but also ensures optimal computational efficiency.

This study contributes to the suicide detection domain by proposing a more accurate and efficient AES. We worked on feature generation using a multi-mode approach to machine learning. Generated features are more effective in achieving significant performance in comparison with original features. We deployed state-of-the-art text preprocessing techniques to clean depression-related text posts. We used several feature extraction techniques such as a bag-of-words (BoW), hashing, and TF-IDF to extract features. We proposed novel features based on probabilities to train machine learning (ML) models. In the end, we deployed machine learning and deep learning models for suicide or non-suicide prediction using depression-related posts. This study has several contributions:

- This study introduces a novel methodology for predicting suicide risk by leveraging depression-related content from social media platforms, employing advanced machine learning techniques.
- In this research, we propose a feature engineering framework that incorporates logistic regression (LR) and support vector machines SVM to calculate prediction probabilities. These probabilities are subsequently integrated into our machine learning models, resulting in improved accuracy and computational efficiency.
- To validate the effectiveness of our approach, comprehensive state-of-the-art comparisons are conducted. We demonstrate how the incorporation of probability-based features enhances model training, setting a new benchmark in the field.
- Our proposed methodology offers a dual advantage. Not only does it enable the generation of high-quality insights from a relatively small dataset, but it also significantly reduces the computational burden, making it suitable for large-scale applications.

This study is further divided into the following sections: Related work is present in 'Related work' and methods used in this study are discussed in 'Material and Methods'.

'Results and Discussion' contains the results of learning models while in the end 'Conclusion and Future Work' presents the conclusion of this study.

## RELATED WORK

AES is trendy in the healthcare domain to make quick and accurate decisions. Different proposed approaches of AES are available for heart disease prediction, COVID-19 prediction, bleeding image classification, stress detection, and emotion detection. All AES uses machine learning, and NLP techniques with different kinds of datasets such as Text, numerical, image, *etc*. For suicide detection, some previous studies used NLP and machine learning techniques with text data to make an accurate AES. In this section, we highlight what kind of work is done by the researcher in the suicide and depression detection domain and what is the gap for other researchers in this domain.

### Deep learning approaches

*Kim et al. (2020)* worked on depression detection using deep learning models and social media data. Frequency-based features they used in their approaches and deployed state-of-the-art LSTM and RNN for depression detection and achieved a 0.99 accuracy score using the RNN model. *Naseem et al. (2022b)* proposed an approach for depression detection using the ordinal classification technique. They worked on depression-level detection and built a new dataset using Reddit posts. The dataset is classified into four target classes using Beck's Depression Inventory and the Depressive Disorder Annotation technique. The hierarchical attention method is optimized to find depression using a soft probability distribution. The proposed approach by the authors outperforms in comparison to the state-of-the-art models. *Zogan et al. (2022)* proposed Multi-Aspect Depression Detection with a Hierarchical Attention Network for depression detection. They extract posts from Twitter to perform experiments. The proposed approach by *Zogan et al. (2022)* achieved a significant 0.895 accuracy score. *Jyothi Prasanth, Dhalia Sweetlin & Sruthi (2022)* proposed a lexicon-based approach for depression detection from Twitter posts. They worked on Neutral Negative scoring algorithms to find the depression and used a Bi-directional long short-term memory algorithm and achieved a 0.90 accuracy score. *Uddin et al. (2022)*, performed experiments for depressive symptoms extraction using deep learning techniques. They used a dataset of young people from Norway collected from an online platform in large volumes. They deployed deep learning models such as LSTM and RNN to identify the text that described the self-perceived symptoms of depression. They used Local Interpretable Model-Agnostic Explanations (LIME) later to extract the symptoms related to depression with significance in comparison stat of the art term frequency technique.

### Machine learning approaches

*Kilaskar et al. (2022)* utilized machine learning techniques to detect depression from text data. They applied LR, SVM, RF, and XGBoost algorithms for this purpose. Experiments were conducted on two datasets, with the highest accuracy of 0.838 achieved by the XGBoost model on dataset 1, and an accuracy of 0.864 achieved by LR on dataset 2. In *Ho et al. (2022)*, the authors experiments were focused on the detection of depressive

disorders. They employed seven machine learning models and incorporated clinical demographic data obtained through functional near-infrared spectroscopy. The study achieved the highest accuracy of 87.98% with a standard deviation of ±8.84% using a multi-modal model. *Aslam et al. (2022)* worked on emotion detection using a dataset of tweets related to cryptocurrencies. They identified people's emotions, including sadness, anger, and surprise, achieving a significant accuracy of 0.99 by combining LSTM and the gated recurrent unit (GRU). In *Govindasamy & Palanichamy (2021)*, an ensemble learning approach was employed for depression detection. They combined two models and proposed a hybrid model named NBTree.

### Suicide prediction using social media data

*Haque et al. (2022)* focuses on detecting suicidal ideation through social network analysis, leveraging NLP and psychology. By analyzing Twitter data, early indications of suicidal thoughts can be identified. A comparative analysis of machine learning and deep learning models is conducted, with the RF model achieving 93% accuracy and a 0.92 F1 score. Training deep learning classifiers with word embedding further enhances model performance, with the BiLSTM model reaching 93.6% accuracy and a 0.93 F1 score. This research contributes to the early recognition of suicidal signs on social media. They introduces a multi-faceted approach to suicide risk detection using deep learning and traditional machine learning models. They analyze social media posts to predict suicide attempts within 30 days and six months. Handcrafted features based on suicide theories and emotional cues are incorporated. Traditional machine learning models outperform the baseline, achieving an F1 score of 0.741 for the 30-day prediction. The deep learning method, however, outperforms the baseline, with an F1 score of 0.737 for the six-month prediction, highlighting its effectiveness in this context. *Ophir et al. (2020)* addresses the challenging task of suicide risk detection using ANN models and Facebook data. The research significantly improves prediction accuracy by employing a multi-task model (MTM) that integrates various risk factors. Importantly, the models rely on a range of text features rather than explicit suicide-related content. This work highlights the potential of machine learning in enhancing suicide risk predictions and advancing practical detection tools. Similarly, *Tadesse et al. (2019)* explores the impact of suicide ideation in social media language. The research focuses on the early detection of suicidal posts on Reddit, employing a combined long short-term memory convolutional neural network (LSTM-CNN) model. Results show that this model, coupled with word embedding techniques, achieves the best classification accuracy, underlining the power of deep learning in assessing suicide risk in text classification tasks.

Table 1 presents a summary of the literature review. Our analysis reveals that previous studies have encountered limitations due to the infrequent use of feature engineering techniques. This is primarily attributed to the fact that the available datasets are often either extensive or small in size and exhibit imbalanced target class ratios. In such scenarios, the extraction of important features becomes crucial for effective model training. Employing the entire feature set, even those that are unimportant, can introduce unnecessary complexity during model training. Consequently, our study places a distinct emphasis on feature

**Table 1  Summary of related work.**

| Ref. | Year | Approach | Data Type | Model | Findings | Limitations |
|---|---|---|---|---|---|---|
| *Burke, Ammerman & Jacobucci (2019)* | 2019 | DL | Text | MLP, SVM, NB | MLP outperform all other used stat of the arts models | Limited dataset for training of models |
| *AlSagri & Ykhlef (2020)* | 2020 | ML | Text | SVM, DT, NB | SVM gives high accuracy in compassion all used stat of the arts models | Cannot avoid over-fit data |
| *Kim et al. (2020)* | 2020 | DL | Text | CNN, XGBoost | CNN high accuracy in comparison all other used stat of the arts models | Limited Scope |
| *Souza Filho et al. (2021)* | 2021 | ML | Text | DT, SVM, RF, GB, LR | RF provides high accuracy | User confession are only the source the evaluate the proposed approach |
| *Govindasamy & Palanichamy (2021)* | 2021 | ML | Text | NB, Hybrid NBTree | Hybrid model NBTree outperform | More computational Cost as compared to individual model. |
| *Aydemir et al. (2021)* | 2021 | ML | EEG Signal | Weighted kNN and Quadratic SVM | KNN outperform in terms of accuracy and efficiency | Can work well for small datasets |
| *Kilaskar et al. (2022)* | 2022 | ML | Text | LR, SVM, RF and XGBoost | XGBoost and LR perform well | Imbalanced dataset |
| *Jyothi Prasanth, Dhalia Sweetlin & Sruthi (2022)* | 2022 | DL | Text | Lexicon technique, Bi-LSTM | Bi-LSTM outperform with significant accuracy | High computational cost as compared to machine learning models. |
| *Sarkar, Singh & Chakraborty (2022)* | 2022 | ML | EEG Signal | LR, SVM, LSTM, RNN, CNN | LR and SVM perform significantly to achieved good results. | Sparse feature space. |

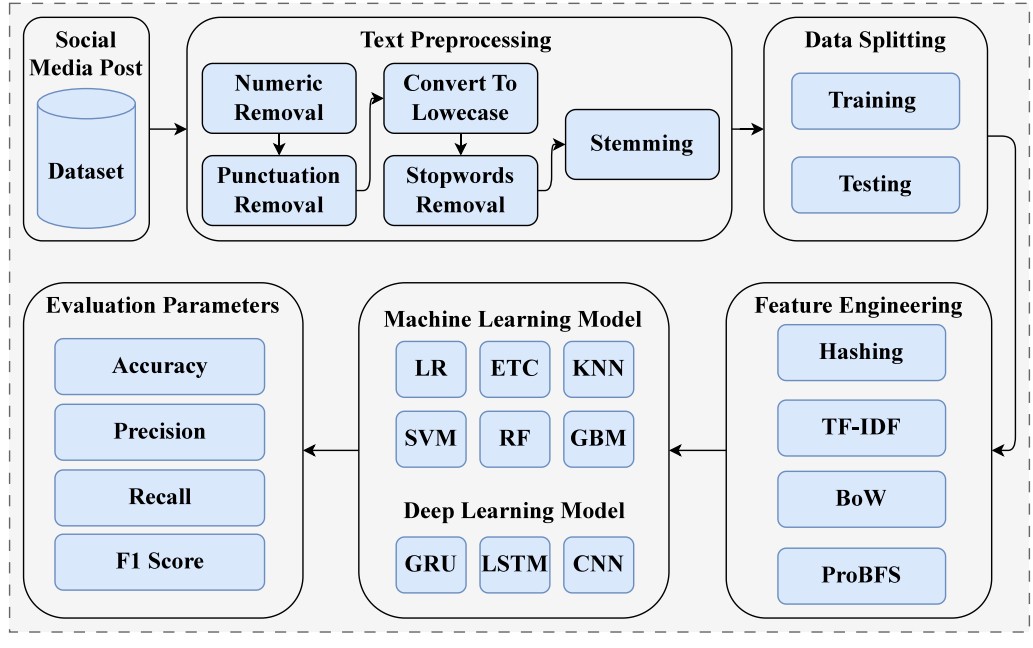

**Figure 1** Block diagram of overall process.

engineering, opting for simplicity over complex models This approach not only leads to notable accuracy improvements but also reduces computational costs to a minimum.

# MATERIAL AND METHODS

This study introduces an Automated Early Warning System (AES) for suicide prediction using social media content. The experimentation was conducted on a machine equipped with an Intel Core i7 7th generation processor, Windows operating system, 16 GB of RAM, and a 1 TB SSD. We implemented the proposed approach using Jupyter Notebook, incorporating NLTK, sci-kit-learn, and the TensorFlow framework. The AES approach is illustrated in Fig. 1

First, we acquired a dataset from the public repository Kaggle (*KOMATI, 2023*). This dataset consists of text data and two target classes (see 'Dataset'). The dataset used comprises social media posts containing a substantial amount of raw information that is not intended for training machine learning models for suicide prediction. To address this issue, we deployed several text preprocessing techniques, including numeric removal, punctuation removal, conversion to lowercase, stop-word removal, and stemming techniques. We split the dataset into training and testing sets with an 80:20 ratio. For model training, 80% of the dataset was used, and for testing, 20% of the dataset was utilized. After data splitting, we applied feature extraction techniques to extract features from text data, such as Bag of Words (BoW), TF-IDF, and Hashing. We also introduced a novel feature generation approach named ProBFS, which consists of two models: SVM and LR. This technique generates a new feature set for machine learning model training, which is more efficient compared to other used features. In the end, we deployed machine learning and deep

**Table 2  Dataset variables.**

| Variable | Description |
| --- | --- |
| No. | ID no. of sample |
| Text | Text posts contain suicide and non-suicide-related content. |
| Class | Target (Suicide—Non-suicide) |

**Table 3  Dataset statistics.**

| Target | Description |
| --- | --- |
| Suicide | 116,037 |
| Non-Suicide | 116,037 |
| Total | 232,074 |
| NoW in small post | 1 |
| NoW in large post | 5,850 |
| Avg. NoW | 58 |

**Table 4  Sample of dataset.**

| No. | Text | Class |
| --- | --- | --- |
| 1 | Ex Wife Threatening SuicideRecently I left my wife for good because she has cheated on me twice and lied to me so much that I have decided to refuse to go back to her. As .... | Suicide |
| 2 | Am I weird I don't get affected by compliments if it's coming from someone I know irl but I feel really good when internet strangers do it | Non-suicide |
| 3 | Finally 2020 is almost over... So I can never hear ''2020 has been a bad year'' ever again. I swear to fucking God it's so annoying | Non-suicide |
| 4 | I need helpjust help me im crying so hard | Suicide |

learning models with their best hyperparameter settings. We evaluated all models in terms of accuracy, precision, recall, and F1 score. Confusion matrix values were also used for model evaluation.

## Dataset

This study used a text dataset for experiments named ''Suicide and Depression Detection'' which consists of social media posts related to suicide and depression. This dataset is publicly available on Kaggle (*KOMATI, 2023*). The dataset contains posts extracted from the SuicideWatch and Depression subreddits of the Reddit platform. Pushshift API is used to extract the dataset and categorize the posts into two classes: Suicide and Non-Suicide posts. The posts extracted from SuicideWatch were created by users from Dec 16, 2008, to Jan 2, 2021, while Depression posts were collected from Jan 1, 2009, to Jan 2, 2021. Attributes of the dataset are shown in Table 2 and a sample of the dataset is shown in Tables 3 and 4 shows the dataset statistics, including the number of words (NoW) in the largest and smallest posts, as well as the count of each target class. Figure 2 shows word-clouds for both target classes suicide and non-suicide.

## Preprocessing

This study used text data in its raw form containing lots of meaningless information such as numbers, punctuation marks, stopwords, *etc.* We used several techniques to clean the data to make the feature set worthy such as punctuation removal, number removal, conversion to lowercase, stopwords removal, and stemming. Preprocessing techniques will remove the meaningless information from data and make the feature set less complex for learning models (*Khan et al., 2021*). We have constructed a preprocessing techniques pipeline, through which we pass our data to obtain preprocessed text. We utilize this preprocessed text in our overall approach. We used a natural language toolkit (NLTK) to perform text preprocessing steps, as shown in Fig. 3.

- **Punctuation removal:** Punctuation marks are the important parts of the sentences to clearly understand the meaning but for machine learning models these punctuation marks are not meaningful because they might be repeated in both target sentences such as punctuation signs (.;") can be in each sample of the dataset. So it can't be helpful to distinguish between suicide and non-suicide posts. We removed punctuation marks using regular expressions.
- **Number removal:** We removed the numbers from text data because they are not meaningful in the text classification task. Numbers have low frequency in sentences so they can't be useful in the feature set for model training. We removed numbers for text using regular expressions.
- **Convert to lowercase:** Machine learning techniques are case sensitive such as "data" and "Data" are the same in meaning but will be different features for model training. Convert to lowercase technique can help to reduce this complexity in the feature set by converting all characters into lowercase. We used the Python function tolower() to convert each character into lowercase.
- **Stopword removal:** Some words are not meant for machine learning models in text such as is, the, and, we, he, *etc.* These words are known as stopwords and we remove them from the feature set to make it simple and avoid complexity.

## Feature engineering

We used three feature extraction techniques in this study which are BoW, TF-IDF, and hashing. BoW and TF-IDF based on terms frequency criteria and hashing technique used a hash trick to map a string into numeric form. All feature extraction techniques are discussed below:

### BoW

Bag-of-words (BoW) is the simplest technique to map the text string into the numeric matrix. BoW finds the frequency of each word in a document and then converts them into a feature matrix. It's very easy to interpret, implement, and understand. It takes text data as input and returns a numeric feature set as output which will be a very simple feature set to train learning models. We used countvectorizer library to implement the BoW technique. Table 5 shows the results of the BoW technique on sample data.

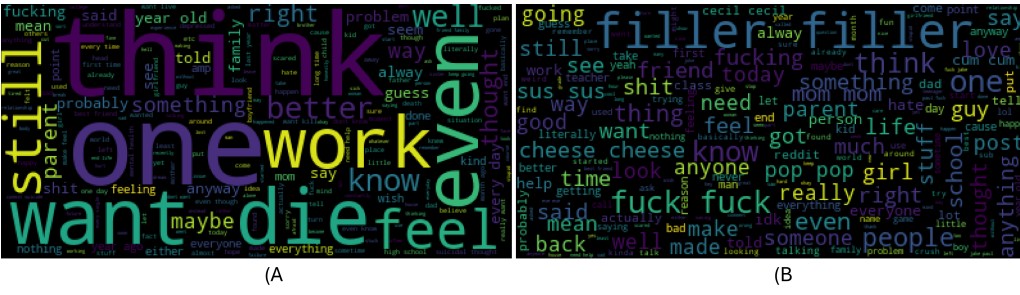

(A                                                    (B

**Figure 2   Word-clouds for suicide and non-suicide classes.**

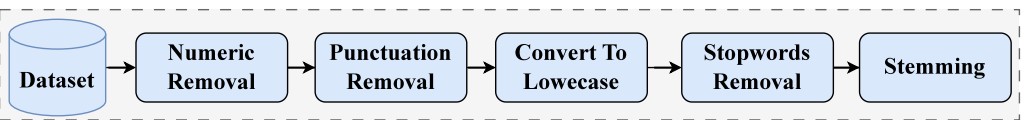

**Figure 3   Preprocessing techniques.**

**Table 5   Results of BoW technique on sample data.**

| No. | Annoying | Do | Feel | Good | Internet | It | Its | Really | So | Strangers | Swear | When |
|-----|----------|-----|------|------|----------|-----|-----|--------|-----|-----------|-------|------|
| 0 | 0 | 1 | 1 | 1 | 1 | 1 | 0 | 1 | 0 | 1 | 0 | 1 |
| 1 | 1 | 0 | 0 | 0 | 0 | 0 | 1 | 0 | 1 | 0 | 1 | 0 |

### Term frequency-inverse document frequency

Term frequency-inverse document frequency (TF-IDF) is a weighted feature extraction technique that assigns weight to each feature in the dataset, unlike BoW which consists of only a simple terms count. TF-IDF is a multiplication of TF and IDF. Its feature score is complex as compared to the simple BoW approach but it is significant for model training because of weighting criteria. TF can be defined mathematically as:

$$tf = TF_{t,d} \tag{1}$$

Here, $t$ is term, $d$ is document and $tf$ is term frequency score for term $t$ in document $d$. IDF can be defined mathematically as:

$$idf = log\left(\frac{N}{D_t}\right) \tag{2}$$

Here, $N$ is the number of documents and $D_t$ is the number of documents containing term $t$. Finally, TF-IDF can be calculated as:

$$tf - idf = TF_{t,d} * log\left(\frac{N}{D_t}\right) \tag{3}$$

Table 6 shows the results of the BoW technique on sample data.

**Table 6  Results of TF-IDF technique on sample data.**

| No. | Annoying | Do | Feel | Good | Internet | It | Its | Really | So | Strangers | Swear | When |
|---|---|---|---|---|---|---|---|---|---|---|---|---|
| 0 | 0.0 | 0.353553 | 0.353553 | 0.353553 | 0.353553 | 0.353553 | 0.0 | 0.353553 | 0.0 | 0.353553 | 0.0 | 0.353553 |
| 1 | 0.5 | 0.000000 | 0.000000 | 0.000000 | 0.000000 | 0.000000 | 0.5 | 0.000000 | 0.5 | 0.000000 | 0.5 | 0.000000 |

**Table 7  Results of hashing technique on sample data.**

| No. | 0 | 1 | 2 | 3 | 4 | 5 | 6 | 7 | 8 | 9 | 10 | 11 |
|---|---|---|---|---|---|---|---|---|---|---|---|---|
| 1 | −0.316228 | 0.632456 | 0.0 | −0.316228 | 0.316228 | 0.0 | 0.316228 | 0.316228 | −0.316228 | 0.0 | 0.0 | 0.000000 |
| 2 | 0.408248 | 0.816497 | 0.0 | 0.000000 | 0.000000 | 0.0 | 0.000000 | 0.000000 | 0.000000 | 0.0 | 0.0 | 0.408248 |

### Hashing

Hashing is one of the efficient techniques in NLP to extract the features. This technique will convert text data into a matrix of token occurrences (*Hasan, Islam & Hasan, 2019*). It takes low memory even for large datasets because it doesn't store vocabulary in memory. Hashing is faster in a pickle and unpickle of data. Table 7 shows the results of hashing techniques on sample data.

### Principal component analysis

Principal component analysis (PCA) is a dimensionality reduction technique employed in the context of suicide prediction from social media posts. After extracting features from the text data using techniques such as TF-IDF and BoW hashing, PCA is applied to reduce the dimensionality of the feature space. This process transforms the original high-dimensional feature set into a lower-dimensional space while preserving as much variance as possible. By capturing the most essential information, PCA contributes to enhancing model efficiency and can improve the performance of machine learning algorithms. We employed PCA to reduce the feature dimension to 15,000, which was then utilized for suicide prediction tasks.

### Chi-squared

Chi-squared (Chi2) feature selection is another vital technique employed in the task of suicide prediction from social media content. Its primary objective is to select the most relevant features from the initial feature space. Chi2 evaluates the statistical independence between each feature and the target variable, which, in this context, represents the prediction of suicide or non-suicide. Features with the highest chi2 scores, signifying a strong dependence on the target variable, are retained, while less informative features are discarded. By reducing the feature dimension to 15,000 using chi2, we ensure that the machine learning models are exposed to the most meaningful textual information, thereby leading to more accurate and efficient predictions.

## Machine learning models

In this study, we employed five machine learning models for suicide detection using social media posts: Random Forest (RF), SVM, Gradient Boosting Machine (GBM), LR, and K-nearest neighbor (KNN). These models have been extensively used in the literature

**Table 8  Hyper-parameters setting for machine learning models.**

| Model | Hyperparameters | Tuning range |
|---|---|---|
| RF | n_estimators= 200; max_depth=200 | n_estimators= 10 to 500; max_depth=10 to 500; |
| ETC | n_estimators= 200; max_depth=200 | n_estimators= 10 to 500; max_depth=10 to 500; |
| GBM | n_estimators= 200; max_depth=200, learning_rate=0.8 | n_estimators= 10 to 500; max_depth=10 to 500; learning_rate=0.1 to 0.9 |
| KNN | n_neighbors= 5; weights=uniform | n_neighbors= 1 to 10; weights= {uniform, distance} |
| SVM | kernel= linear; $C = 2.0$ | kernel= {linear, poly,sigmoid}; $C = 1.0$ to 5.0 |
| LR | solver= liblinear; $C = 2.0$ | solver= {liblinear, saga, sag}; $C = 1.0$ to 5.0 |

for suicide detection. We fine-tuned the models by identifying the best hyperparameter settings through a grid search method. The models were tuned within specific parameter ranges, and the optimal hyperparameter settings are presented in Table 8.

### RF

Random Forest (RF) a tree-based model that uses the ensemble learning concept. RF can be used for classification and regression tasks. It can work well on small datasets as well as on large datasets. RF can efficiently handle the over-fitting problem on imbalanced datasets. RF used the number of decision trees to train and make the decision about the final prediction. Each weak learner (decision tree) predicts a target class and then prediction by each decision tree will pass through majority voting criteria. The target class with the most votes will be selected as the final prediction by RF. We used the entropy algorithm to build the decision trees in RF and we can define decision tree mathematically as:

$$Entropy = -\sum_{i=1}^{n} p_i * log(p_i) \tag{4}$$

Here, $n$ is the number of target classes and $p_i$ is the probability of a class. We can define RF mathematically as:

$$rf = mode\{\sum_{i=1}^{N} dt_i\} \tag{5}$$

Here, *mode* represents the function that identifies the most frequently predicted class among the individual decision trees. Each $dt_i$ corresponds to an individual decision tree, and $N$ signifies the total number of decision trees used.

In this study, we employed 300 decision trees in the prediction process. To manage the complexity of decision trees and the RF, we imposed a maximum depth of 200 levels on each decision tree by using the max_depth hyper-parameter.

### Extra trees classifier

The extra trees classifier (ETC), an ensemble machine learning model, builds upon the concept of decision trees. During training, it constructs multiple decision trees, and predictions are generated by aggregating the results from each tree. The key distinction between ETC and RF lies in its introduction of extra randomness. ETC selects random

subsets of features at each split, reducing the risk of overfitting and enhancing the model's robustness. ETC predicts the class label $\hat{y}$ for an input sample $X$ based on a set of decision rules represented by $T$ decision trees:

$$\hat{y} = \text{Mode}(T_1(X), T_2(X), \ldots, T_n(X))$$

Here, $T_i(X)$ represents the prediction of the $i$-th decision tree for input $X$, and $\text{Mode}(\ldots)$ indicates the most frequently predicted class label among all decision trees.

### LR

Logistic regression a statistical model that can be used for the classification of data. LR is best for the binary classification problem as in our study and it's preferred when the dependent variable is categorical (*Lemon et al., 2003*). LR finds the relationship between the dependent and independent variables. It used the sigmoid function to categorize the data and we can define it mathematically as:

$$Output = suicide(1)|Non - suicide(0)$$
$$Hypothesis \geq Z = WX + B$$
$$h\theta(x) = sigmoid(Z)$$
$$sig(t) = \frac{1}{1 + e^{-t}}. \tag{6}$$

Here, if $Z$ will be $+\infty$ then the prediction will be 1 and if it goes to $-\infty$ then the prediction will be 0. We used LR with liblinear value for the solver hyper-parameter. This parameter is used for optimization which is faster as compared to others. We used a multi_class hyper-parameter with the value "ovr" which we selected because of the binary classification problem.

### Ensemble machine learning model

Gradient Boosting Machine is also an ensemble model used for classification and regression problems. GBM used weak learners (decision trees) as base models and trained them sequentially unlike RF to make the final prediction. This sequential training of base models will help to reduce the error rate. The first decision tree error will be the input of the second decision tree followed by the first tree and so on. We can define GBM mathematically as:

$$f(x) = \sum_{b=1}^{B} f^b(x). \tag{7}$$

Here, $b$ is a decision tree and $x$ is a single predictor. We used 300 decision trees under GBM prediction criteria and restricted each tree to max 200 level depth using max_depth hyper-parameter. We also used learning_rate with 0.2 value which finds how much each tree contributes to outcomes.

### SVM

Support vector machine is a linear model that can be used for regression and classification tasks. SVM can perform well for both binary classification problems and multi-class problems. SVM used a hyperplane to categorize the data. It draws multiple hyperplanes in

feature space and the hyperplane which can separate the data with the best margin will be used for classification. We used a linear kernel as a hyper-parameter with SVM because of the linear feature set we used for experiments.

### KNN

K-nearest neighbor (KNN) is the simplest classification algorithm from the machine learning models family. It can be used for classification and regression tasks. KNN finds the distance between neighbors to classify them according to training data (*Zhang et al., 2017*). KNN matches new data with the training data and classifies new data with the most matched target class in training data. Distance between training data and new data can be calculated using Euclidean distance defined as:

$$Euclidean\ distance = \sqrt{\sum_{i=1}^{k}(x_i - y_i)^2}. \tag{8}$$

We used KNN with n_neighbour hyper-parameter. We used this hyper-parameter with the value "3" which means that 3 neighbors will be looked around to find the distance.

## Proposed multi-modal feature generation approach

In this section, we present the novel feature generation approach named probability-based feature set (ProBFS). The model's performance totally depends on the features for training. If the feature set will be correlated to the target classes then the performance of machine learning models will be significant. There can be several challenges in feature engineering such as what should be the size of the feature set or whether is feature set accurately corresponds to the target class or not. ProBFS technique generates a new feature set that will be more worthy as compared to the original feature set which will help to achieve significant results in terms of accuracy and efficiency.

The proposed ProBFS consists of two machine learning models and TF-IDF features. After extracting features from text data using the TF-IDF technique we pass them to the two best performers of this study LR and SVM. These models get trained on the whole dataset and then predict probabilities for the whole dataset. There will be two probabilities by each model for each sample so in this way, we generate four features using two models. The architecture of the proposed ProBFS is shown in Fig. 4 and Algorithm 1.

The used machine learning models in ProBFS are deployed with the same setting as mentioned in subsection 'Machine learning models'. The newly generated set using ProBFS will be small in size which will help to reduce the computational time of machine learning models. The new feature set will be more correlated to the target class because it is based on the probabilities generated by the models. It will help to achieve significant accuracy.

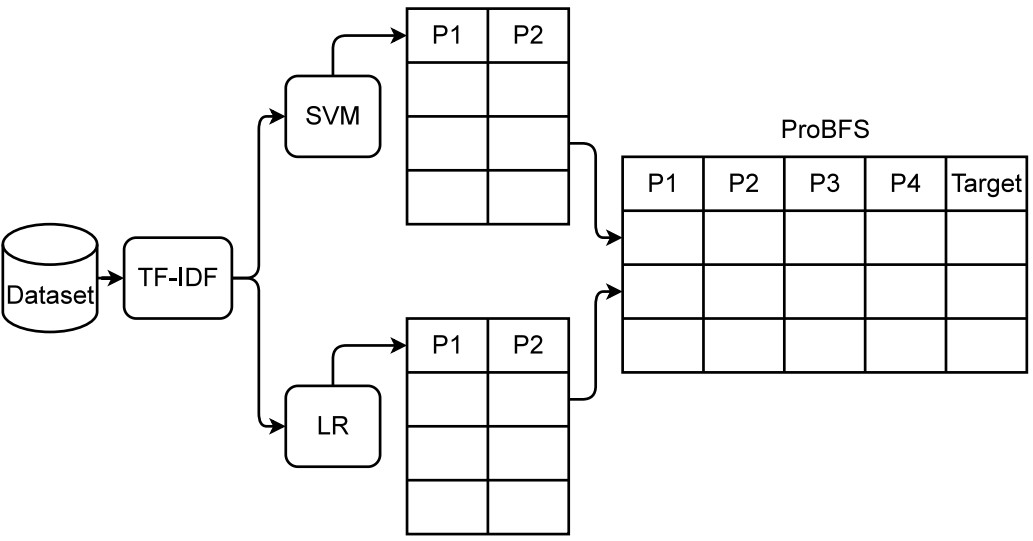

**Figure 4**  **ProBFS architecture.**

---

**Algorithm 1** ProBFS algorithm for feature generation

---

**Input:** Text Posts **Output:** Suicide|Non-Suicide initialization;

    tfidf ⟵ TfidfVectorizor (posts)

    [P1, P2] ⟵ SVM (tfidf)

    [P3, P4] ⟵ LR (tfidf)

    ProBFS ⟵ [P1, P2, P3, P4]

---

# RESULTS AND DISCUSSION

This section presents the results of machine learning and deep learning models for suicide detection in social media posts, aiming to predict suicide or non-suicide targets. We evaluate all models in terms of accuracy, recall, precision, and F1 scores.

## Results of machine learning models

We evaluate the performance of models using three features such as BoW, TF-IDF, hashing and proposed ProBFS. These features convert the text data into numerical form because we can't directly feed the text data to learning models. Table 9, shows the results of machine learning models for depression detection using BoW features. The performance of linear models is good in comparison with tree-based models and others. Linear models such as LR and SVM perform significantly with 0.93 and 0.92 accuracy scores, respectively. While GBM and RF are also good with 0.88 and 0.86 accuracy but not significant in comparison to LR and SVM. KNN is low in accuracy score because of a large feature set in training. KNN performs better when the feature set is small while linear models perform well because of the large feature set. Figure 5 shows the confusion matrix values for each model. LR is on top with more accurate predictions such as it gives 43,021 correct predictions and 3394

**Table 9  Results using the BoW features.**

| Model | Accuracy | Precision | Recall | F1 Score |
|---|---|---|---|---|
| LR | 0.93 | 0.93 | 0.93 | 0.93 |
| GBM | 0.88 | 0.88 | 0.88 | 0.88 |
| RF | 0.86 | 0.86 | 0.86 | 0.86 |
| SVM | 0.92 | 0.92 | 0.92 | 0.92 |
| ETC | 0.77 | 0.80 | 0.77 | 0.76 |
| KNN | 0.77 | 0.78 | 0.77 | 0.77 |

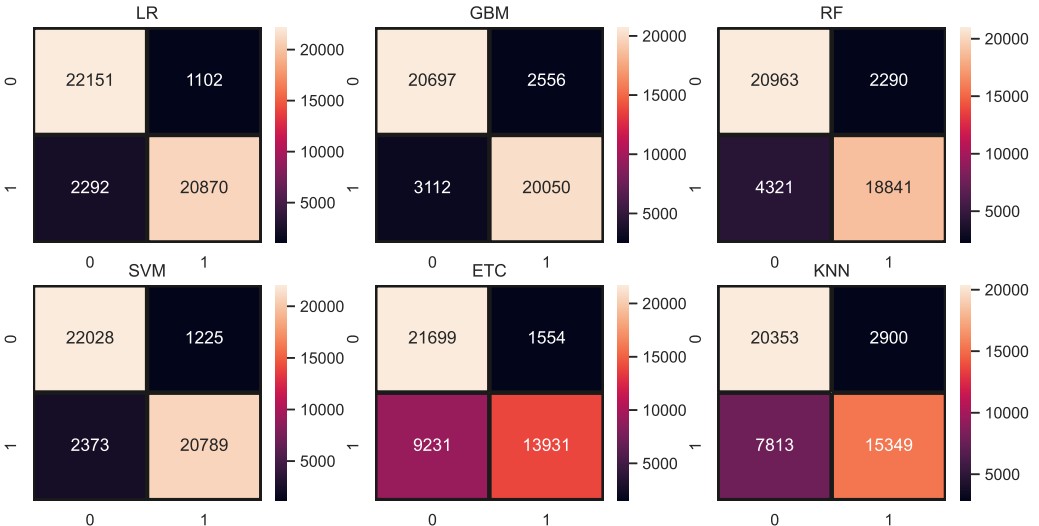

**Figure 5  Models confusion matrices using BoW features.**

wrong predictions out of a total of 46,316 test predictions. KNN is poor in performance because of the high wrong prediction ratio, which is 10,713 out of 46,316 predictions.

Table 10 shows the results of machine learning models using the TF-IDF features. TF-IDF generates more worthy features for model training that's the reason the results of models are somehow improved as compared to results with BoW. LR achieved the highest accuracy of 0.94 with the TF-IDF feature and SVM also performed equally well with a 0.94 accuracy score. This significant performance is because of a large and weighted feature set generated by TF-IDF. Other models such as RF, GBM, ETC, and KNN still in low accuracy scores as compared to linear models. Figure 6 shows the confusion matrix for each model using TF-IDF features. LR gives a total of 43,396 correct predictions and 2,920 wrong predictions out of 46,316 predictions while this time also equal in accuracy score but significant in terms of correct predictions as compared to LR. SVM gives 43,465 correct predictions and 2,851 wrong predictions. The ratio of SVM correct predictions is high as compared to LR.

**Table 10  Results using the TF-IDF features.**

| Model | Accuracy | Precision | Recall | F1 Score |
|---|---|---|---|---|
| LR | 0.94 | 0.94 | 0.94 | 0.94 |
| GBM | 0.88 | 0.88 | 0.88 | 0.88 |
| RF | 0.86 | 0.86 | 0.86 | 0.86 |
| SVM | 0.94 | 0.94 | 0.94 | 0.94 |
| ETC | 0.83 | 0.84 | 0.83 | 0.83 |
| KNN | 0.52 | 0.65 | 0.52 | 0.39 |

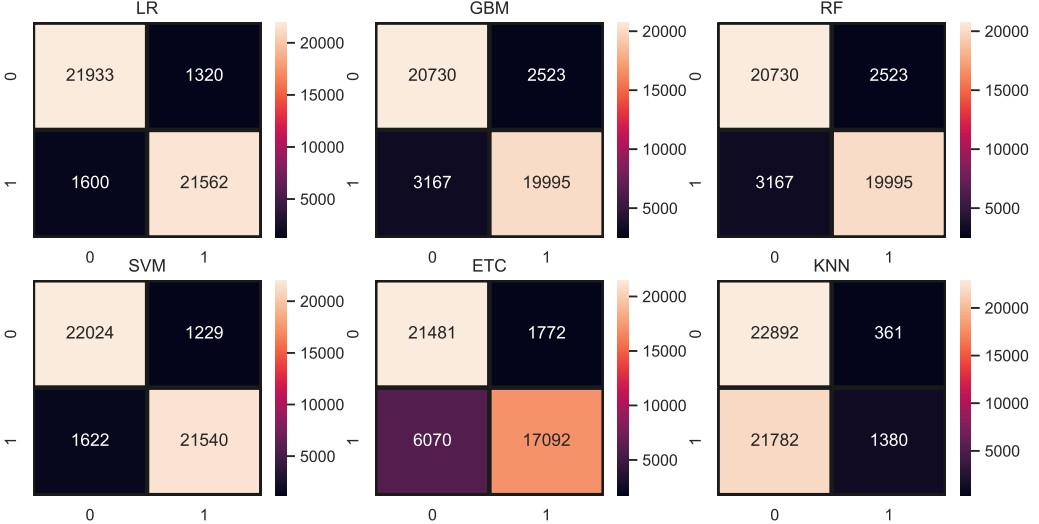

**Figure 6  Models confusion matrices using TF-IDF features.**

Table 11 shows the results of machine learning models using hashing features. SVM is higher in accuracy score as compared to LR with hashing features. SVM achieved a 0.94 accuracy score while LR is just behind the SVM with a 0.93 accuracy score. Other models are still low in accuracy score just because linear feature set. Using all feature set tree-based models and KNN is poor in performance and to improve their performance we have to generate a more linearly separable feature set. Figure 7 shows the confusion matrix for each model using the hashing features. SVM gives the highest ratio for correct predictions which is 43,391 out of 46,316 and gives 2,925 wrong predictions. The lowest wrong predictions ratio is given by the SVM which is 2,851 out of 46,316 predictions using the TF-IDF feature in comparison with BoW and hashing features.

Tree-based models and KNN are not good in terms of accuracy score so we try to reduce the feature set because tree-based models and KNN models perform well when the feature set is small. So, we also experiment by reducing the feature set using Chi-Squared (Chi2) and principal component analysis (PCA) techniques. We select 5,000 features using each feature reduction technique. PCA concludes the directions of maximum variance in high-dimensional feature space and fits it into a new feature space with fewer dimensions

**Table 11   Results using the hashing features.**

| Model | Accuracy | Precision | Recall | F1 Score |
|-------|----------|-----------|--------|----------|
| LR | 0.93 | 0.93 | 0.93 | 0.93 |
| GBM | 0.88 | 0.88 | 0.88 | 0.88 |
| RF | 0.83 | 0.83 | 0.82 | 0.82 |
| SVM | 0.94 | 0.94 | 0.94 | 0.94 |
| ETC | 0.84 | 0.84 | 0.83 | 0.83 |
| KNN | 0.62 | 0.71 | 0.62 | 0.58 |

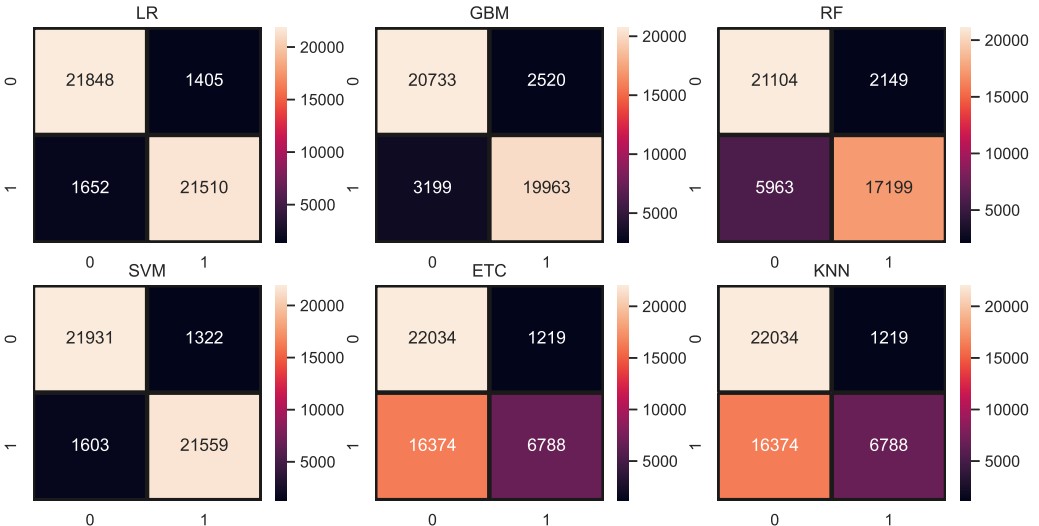

**Figure 7   Models confusion matrices using hashing features.**

as compared to the original. On the other hand, Chi2 finds the relationship between non-negative stats and target class to reduce the feature space. Table 12 shows the results of machine learning models using each feature reduction technique. With Chi2 LR achieved a 0.93 accuracy score while SVM achieved only a 0.92 accuracy score. Using PCA both LR and SVM achieved a 0.92 accuracy score. These results show that linear models didn't improve the accuracy by reducing the features however there is a 1% decrease in accuracy. KNN improved its accuracy with a small feature set as compared to the original. Figure 8 shows the confusion metrics of machine learning models using PCA and Chi2 features.

Table 13 shows the results of machine learning models using the proposed ProBFS. The ProBFS is small in size but more correlated to the target class that's the reason it improved the performance of the learning mode. SVM is significant with the proposed feature set as it achieved the 0.96 accuracy score and also outperformed in terms of all other evaluation parameters. LR, RF, ETC, and KNN all perform well with a 0.95 accuracy score. Models improved 2% accuracy using the proposed feature set as compared to the other used features. KNN achieved its highest accuracy in the study which is 0.95 because the feature set is too small which is more suitable for it as well as for linear models.

**Table 12  Results using the Chi2 features.**

| Feature | Model | Accuracy | Precision | Recall | F1 Score |
|---------|-------|----------|-----------|--------|----------|
| Chi2 | LR | 0.93 | 0.93 | 0.93 | 0.93 |
| | GBM | 0.87 | 0.87 | 0.87 | 0.87 |
| | RF | 0.88 | 0.88 | 0.88 | 0.88 |
| | SVM | 0.92 | 0.92 | 0.92 | 0.92 |
| | ETC | 0.79 | 0.81 | 0.79 | 0.78 |
| | KNN | 0.75 | 0.75 | 0.75 | 0.75 |
| PCA | LR | 0.92 | 0.92 | 0.92 | 0.92 |
| | GBM | 0.87 | 0.87 | 0.87 | 0.87 |
| | RF | 0.86 | 0.86 | 0.86 | 0.86 |
| | SVM | 0.92 | 0.92 | 0.92 | 0.92 |
| | ETC | 0.79 | 0.79 | 0.79 | 0.79 |
| | KNN | 0.77 | 0.77 | 0.77 | 0.77 |

**Table 13  Results using the ProBFS features.**

| Model | Accuracy | Precision | Recall | F1 Score |
|-------|----------|-----------|--------|----------|
| LR | 0.95 | 0.95 | 0.95 | 0.95 |
| GBM | 0.93 | 0.93 | 0.93 | 0.93 |
| RF | 0.95 | 0.95 | 0.95 | 0.95 |
| SVM | 0.96 | 0.96 | 0.96 | 0.96 |
| ETC | 0.95 | 0.95 | 0.95 | 0.95 |
| KNN | 0.95 | 0.95 | 0.95 | 0.95 |

Figure 9 shows the confusion metrics for each model using the proposed feature set. SVM gives only 2,078 wrong predictions and 44,238 correct prediction predictions out of 46,316 test predictions. This is the highest correct prediction ratio we achieved using machine learning models. KNN gives 43,823 correct predictions which are the highest as compared to previous results of KNN. GBM is poor in performance in this case with 2,733 wrong predictions because of the large size of data but still better as compared to model performance using the original feature on other used features. Figure 10 models the accuracy comparison using BoW, TF-IDF, hashing, and ProBFS techniques.

## Deep learning models results

We also used deep learning models in comparison with machine learning models. Deep learning models such as LSTM, CNN, and GRU are used with their state-of-the-art architectures. LSTM and GRU are the recurrent applications that are recommended for text classification while CNN is specially designed for image processing but it also has text classification applications.

We cannot directly feed text data to the learning models, so we need to convert it into a numeric sequence. To accomplish this, we utilized the TensorFlow library to convert the text data into sequences, followed by sequence padding. Once the data was sequenced, we incorporated an embedding layer for each model, which is defined by three crucial

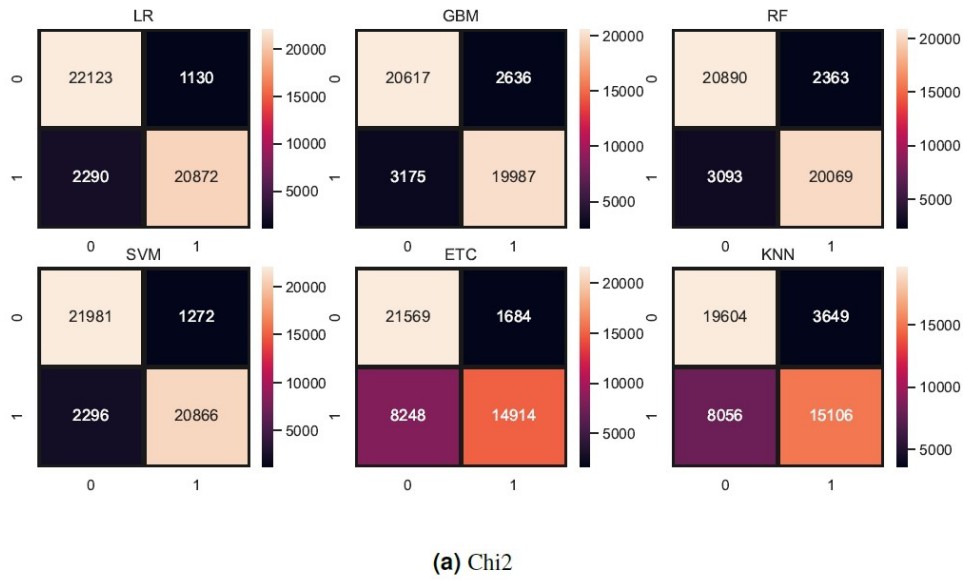

**(a)** Chi2

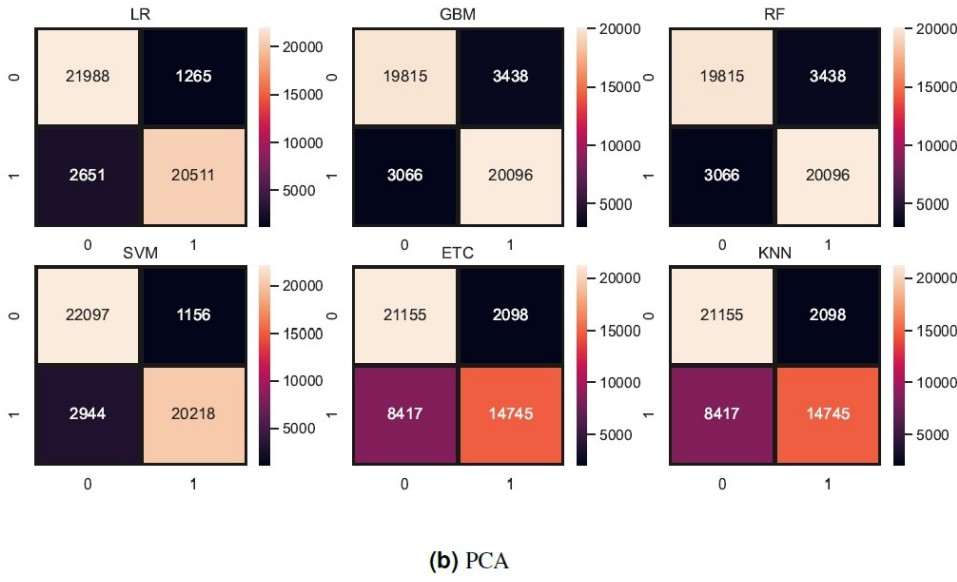

**(b)** PCA

**Figure 8  (A–B) Models confusion matrices using feature reduction techniques.**

parameters: vocabulary size, output dimension, and input length. In our experiment, the vocabulary size was set to 5,000, indicating that the models can accept input values up to 5,000 in the input feature set. The output dimension parameter determines the size of the output produced by the embedding layer.

LSTM model will take the output of the embedding layer as input. LSTM models consist of an LSTM layer with 100 units, and a 0.5 dropout rate which will drop 50% of neurons from models during training to reduce the complexity of models (*Aslam et al., 2022*). CNN model also takes the output of the embedding layer as input. 1D CNN layer takes this

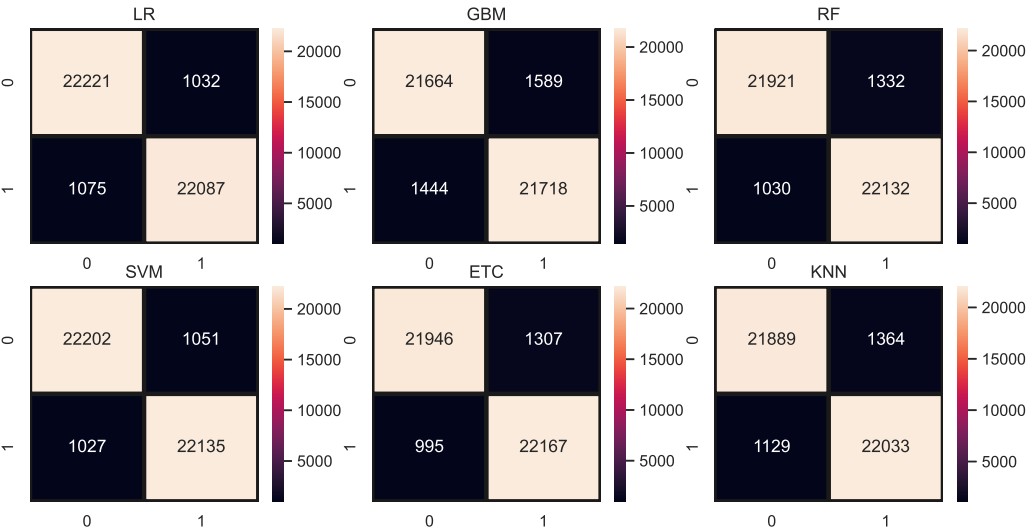

**Figure 9** Models confusion matrices using proposed ProBFS.

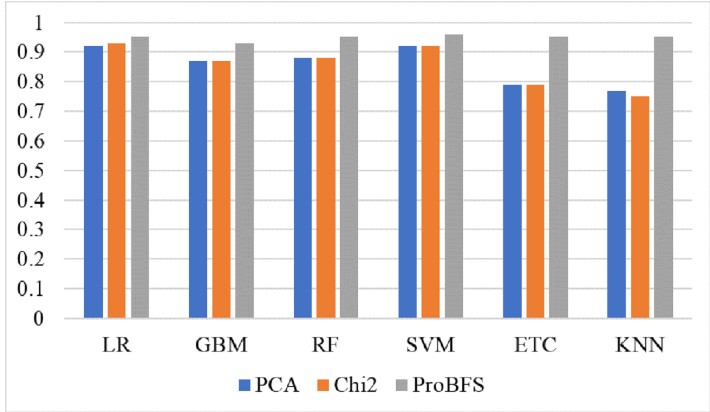

**Figure 10** Models accuracy comparison using PCA, Chi2 and ProBFS techniques.

output as input to proceed with it. This 1D CNN layer consists of 128 filters and 3 by 3 kernel size and ReLU activation function. Max-pooling layer with a 3 by 3 pool size is followed by the 1D CNN layer. After the max-pooling layer, there is an activation layer with the ReLU activation function, a dropout layer with a 0.5 dropout rate, and then a Flatten layer to convert three-dimensional data into 1-dimensional data. In the end, data will pass to a dense layer with 32 neurons. Similarly, like LSTM and CNN, GRU will also take embedding layer output as input. GRU model consists of a dropout layer with a 0.5 drop rate. After this, we used a GRU layer with 100 units followed by a dropout layer with a 0.5 dropout rate. This dropout layer is followed by a dense layer consisting of 32 neurons.

In the end, all models are compiled using binary_crossentropy loss function and Adam optimizer. Models are fitted with 100 epochs and a 128 batch size value because smaller

**Table 14 Architecture of deep learning models.**

| LSTM | GRU | CNN |
|---|---|---|
| Embedding(5000,100, input_length=..) | Embedding(5000,100, input_length=..) | Embedding(5000,100, input_length=..) |
| Dropout(0.5) | Dropout(0.5) | Conv1D(128, 3, activation='relu') |
| LSTM(100) | GRU(100) | MaxPooling1D(pool_size=3) |
| Dropout(0.5) | Dropout(0.5) | Activation('relu') |
| Dense(32) | Dense(32) | Dropout(rate=0.5) |
| Dense(2, activation='softmax') | Dense(2, activation='softmax') | Flatten() |
|  |  | Dense(32) |
|  |  | Dense(2, activation='softmax') |

loss='binary_crossentropy', optimizer='adam',
epochs=100, batch_size=128

**Table 15 Results using the deep learning models.**

| Feature | Model | Accuracy | Precision | Recall | F1 Score |
|---|---|---|---|---|---|
| Original | LSTM | 0.91 | 0.91 | 0.91 | 0.91 |
|  | CNN | 0.90 | 0.90 | 0.90 | 0.90 |
|  | GRU | 0.92 | 0.92 | 0.92 | 0.92 |
| ProBFS | LSTM | 0.55 | 0.55 | 0.55 | 0.55 |
|  | CNN | 0.45 | 0.45 | 0.45 | 0.45 |
|  | GRU | 0.57 | 0.57 | 0.57 | 0.57 |

batch sizes (*e.g.*, 32 or 64) can introduce more noise into the training process, stemming from individual samples (*McCandlish et al., 2018*). Table 14 shows the architecture of deep learning models.

Table 15 shows the results of deep learning models using original text features and ProBFS. These results show that models didn't perform well with ProBFS as well as with Original features. GRU achieved a 0.92 accuracy score with the original features and 0.57 with the ProBFS. Figure 11 shows the comparison between BoW, TF-IDF, Hashing, and ProBFS for the machine learning model's performance.

**Limitation:** The results with deep learning models show the limitation of ProBFS in this study. Our proposed approach ProBFS generates a new feature set using the prediction probability by the machine learning models and results in the form of a small feature set. The new small feature set is not enough to train deep learning models to achieve significant results. It can be better only for machine learning models. We include this limitation in our future work.

## Computational time

The model's efficiency is also important as well as the accuracy. We measure the computational cost in terms of training and prediction time to show the significance of the proposed approach. Table 16 shows computational time in seconds for each model with each feature. We can see the significance of the proposed approach in terms of efficiency. All models take very little time for computation using ProBFS as compared to

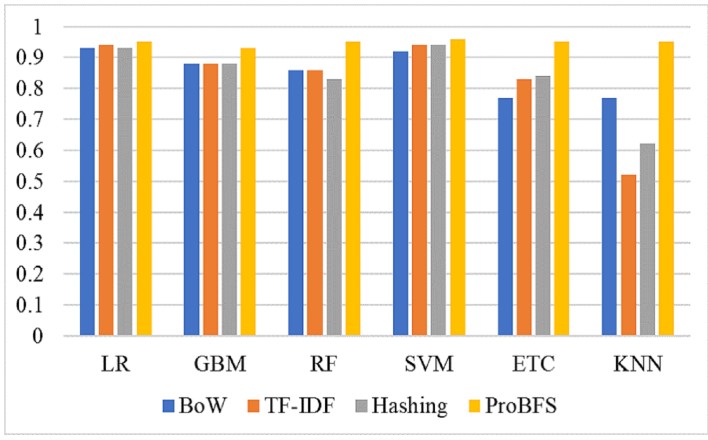

**Figure 11** Models accuracy comparison using BoW, TF-IDF, Hashing, and ProBFS techniques.

**Table 16    Computational time (seconds) of each machine learning model.**

| Model | BoW | TF-IDF | Hashing | ProBFS |
|-------|-----|--------|---------|--------|
| LR | 115.45 | 111.02 | 1,345.74 | 7.73 |
| GBM | 2,531.42 | 2,487.32 | 3,178.01 | 348.54 |
| RF | 1,414.02 | 1,248.85 | 1,510.91 | 272.34 |
| SVM | 66.06 | 87.45 | 145.32 | 5.63 |
| ETC | 1,224.52 | 1,294.86 | 1,666.54 | 38.15 |
| KNN | 1,024.98 | 905.40 | 1,224.38 | 5.98 |

other features because ProBFS consists of only four correlated features while others are huge in numbers. SVM takes only 5 s and gives significant 0.96 accuracies.

## K fold cross validation results

To demonstrate the significance of the proposed ProBFS technique, we conducted experiments using 10-fold cross-validation. We deployed machine learning models with the proposed feature set. The models' performance was very consistent with 10-fold cross-validation. SVM achieved a significant mean accuracy score of 0.96 with a standard deviation (SD) of 0.01. Similarly, ETC also demonstrated a significant mean accuracy score of 0.95. These results validate the proposed approach and indicate no signs of model overfitting. The results of K-fold cross-validation are shown in Table 17.

## Comparison with previous approaches

To show the significance of the proposed approach we have done a comparison with previous methods. We deployed proposed approaches from recent studies that show effective results in text classification and suicide detection such as *Jyothi Prasanth, Dhalia Sweetlin & Sruthi (2022)*, who used Bi-LSTM for depression detection using Twitter data. *Aslam et al. (2022)* proposed LSTM-GRU ensemble models for emotion detection from text tweets. Similarly, another study worked on depression detection using a tweets dataset and used Bi-LSTM for it. *Amanat et al. (2022)* used deep learning models RNN and LSTM

**Table 17  K-fold cross-validation results using the ProBFS features.**

| Model | Mean | SD |
|---|---|---|
| LR | 0.94 | +/−0.09 |
| GBM | 0.93 | +/−0.00 |
| RF | 0.94 | +/−0.02 |
| SVM | 0.96 | +/−0.01 |
| ETC | 0.95 | +/−0.01 |
| KNN | 0.93 | +/−0.13 |

**Table 18  Comparison results with previous approaches.**

| Ref. | Year | Model | Accuracy | F1 score |
|---|---|---|---|---|
| *Amanat et al. (2022)* | 2022 | RNN-LSTM | 0.91 | 0.90 |
| *Jyothi Prasanth, Dhalia Sweetlin & Sruthi (2022)* | 2022 | Bi-LSTM | 0.90 | 0.89 |
| *Aslam et al. (2022)* | 2022 | LSTM-GRU | 0.92 | 0.92 |
| *Motade et al. (2022)* | 2022 | SVM | 0.93 | 0.93 |
| This study | 2022 | SVM | 0.96 | 0.96 |

for depression detection from textual data. *Motade et al. (2022)* experimented with suicide detection using the SVM model. For a fair comparison, we deployed all these approaches on our used dataset. Table 18 shows the comparison results and according to the results in our study, SVM achieved significant results because of the proposed ProBFS.

# CONCLUSION AND FUTURE WORK

An ASE is deployed in this study for suicide detection to predict suicide or non-suicide posts. We used machine learning and deep learning models with several feature engineering techniques. Additionally, we proposed a novel feature generation technique, ProBFS, for model training. SVM achieved a significant accuracy score of 0.96 using the proposed ProBFS, with low computational cost. Through extensive experiments and analysis, we draw several conclusions in this study.

First, we concluded that linear models LR and SVM can perform well on both large feature sets and small linearly separable feature sets. In this study, we achieved the highest results on both large and small feature sets with SVM. Second, we found that KNN and tree-based models RF, ETC can excel when dealing with small feature sets. KNN improved its performance from 0.79 to 0.95 when we used the proposed feature set for training. Third, we determined that a small feature set can work well for machine learning models but not for deep learning models, even if it is linearly separable. The proposed features, ProBFS, helped improve the performance of machine learning models. However, for deep learning models, they were not suitable. Deep learning models experienced a drop in accuracy scores with the proposed feature set, indicating that a small feature set is not suitable for neural networks. In the end, we also concluded that probability-based small features can help reduce computational costs, and make models more efficient while maintaining accuracy.

Despite all these advantages, this study also has some limitations, such as ProBFS's size depending on the number of target classes and the number of machine learning models. If the number of target classes is smaller and there are fewer models, as in this study, the new feature set may be small and insufficient for training deep learning models. In future work, we will address this limitation and strive to create a feature set that works well for both machine learning and deep learning models.

### Funding
This study was supported by the National Natural Science Foundation of China (Grant No.41902065), the Third Xinjiang Scientific Expedition Program (2022xjkk1303) and the China Geological Survey Program (ZD20220126). The funders had no role in study design, data collection and analysis, decision to publish, or preparation of the manuscript.

### Grant Disclosures
The following grant information was disclosed by the authors:
National Natural Science Foundation of China: 41902065.
Third Xinjiang Scientific Expedition Program: 2022xjkk1303.
China Geological Survey Program: ZD20220126.

### Competing Interests
Ting Ding is employed by Urumqi Comprehensive Survey Center on Natural Resources. Tonghui Qu is employed by Hangzhou Hikvision Digital Technology Co. Jiayan Zhang is employed by Nanomega CryoemAI Co. The authors declare there are no competing interests.

### Author Contributions
- Ting Ding conceived and designed the experiments, analyzed the data, prepared figures and/or tables, authored or reviewed drafts of the article, and approved the final draft.
- Tonghui Qu analyzed the data, authored or reviewed drafts of the article, and approved the final draft.
- Zongliang Zou performed the experiments, analyzed the data, performed the computation work, prepared figures and/or tables, authored or reviewed drafts of the article, and approved the final draft.
- Cheng Ding conceived and designed the experiments, performed the experiments, analyzed the data, performed the computation work, prepared figures and/or tables, authored or reviewed drafts of the article, and approved the final draft.

### Data Availability
  The raw code is available in the Supplementary File.
  The datasets analyzed during the current study are available at Kaggle: https://www.kaggle.com/datasets/nikhileswarkomati/suicide-watch.

## Supplemental Information

Supplemental information for this article can be found online at http://dx.doi.org/10.7717/peerj-cs.2301#supplemental-information.

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
