# Peer review of "A novel multi-model feature generation technique for suicide detection"

_PeerJ Computer Science, doi:10.7717/peerj-cs.2301_

## Round 0.1 · original submission · Major Revisions

Dear authors,

Thank you for your submission. Your article has not been recommended for publication in its current form. However, we do encourage you to address the concerns and criticisms of the reviewers and resubmit your article once you have updated it accordingly.

Best wishes,

**Language Note:** The review process has identified that the English language must be improved. PeerJ can provide language editing services - please contact us at [email protected] for pricing (be sure to provide your manuscript number and title). Alternatively, you should make your own arrangements to improve the language quality and provide details in your response letter. – PeerJ Staff

Reviewer 1 ·

Basic reporting

1、The article includes sufficient introduction and background information; however, in the introduction section, the author separately discusses the background and their own methodology without analyzing the existing problems. There is some analysis of the problems in the later related work section. It is recommended to briefly summarize the identified issues in the introduction section. Additionally, in the introduction part, when summarizing the contributions of the article, the author's statements are somewhat vague, such as "This study proposed an approach for suicide prediction using depression-related social media posts and machine learning methods." It is suggested to provide a concise summary of the proposed methods.

2、In the "Materials and Methods" section, it is recommended to adjust the orientation of the text in Figures 1 and 3 within the preprocessing techniques section to enhance the readability of the images.

3、In section 3.4.1, it is recommended to provide an explanation for "mode{}" in Equation (5).

Experimental design

1、The dataset provided in the paper is mentioned as "suicide and depression detection (https://www.kaggle.com/datasets/nikhileswarkomati/suicide-watch)." However, the actual dataset used in the code appears to be related to "human stress detection during sleep (https://www.kaggle.com/datasets/laavanya/human-stress-detection-in-and-through-sleep)." There seems to be an inconsistency between the dataset mentioned and the one used in the code. It is advisable for the authors to verify and clarify this discrepancy.

Validity of the findings

no comment

Cite this review as

Reviewer 2 ·

Basic reporting

My considerations are in the attached PDF.

Experimental design

My considerations are in the attached PDF.

Validity of the findings

My considerations are in the attached PDF.

Annotated reviews are not available for download in order to protect the identity of reviewers who chose to remain anonymous.
Cite this review as

---

## Round 0.2 · accepted · Accept

Dear authors,

Thank you for clearly addressing all the reviewers' comments. I confirm that the quality of your paper has been improved. The paper now appears to be ready for publication in light of this revision.

Best wishes,

Reviewer 1 ·

Basic reporting

no comment

Experimental design

no comment

Validity of the findings

no comment

Additional comments

There is an incorrect reference in Table 18, marked as a "?".

Cite this review as